# Optical Detection Methods for High-Throughput Fluorescent Droplet Microflow Cytometry

**DOI:** 10.3390/mi12030345

**Published:** 2021-03-23

**Authors:** Kaiser Pärnamets, Tamas Pardy, Ants Koel, Toomas Rang, Ott Scheler, Yannick Le Moullec, Fariha Afrin

**Affiliations:** 1Thomas Johann Seebeck Department of Electronics, Tallinn University of Technology, 19086 Tallinn, Estonia; ants.koel@taltech.ee (A.K.); toomas.rang@taltech.ee (T.R.); yannick.lemoullec@taltech.ee (Y.L.M.); fariha.afrin@taltech.ee (F.A.); 2Department of Chemistry and Biotechnology, Tallinn University of Technology, 19086 Tallinn, Estonia; tamas.pardy@taltech.ee (T.P.); ott.scheler@taltech.ee (O.S.)

**Keywords:** droplet microfluidics, optical sensors, light sources, microflow cytometry

## Abstract

High-throughput microflow cytometry has become a focal point of research in recent years. In particular, droplet microflow cytometry (DMFC) enables the analysis of cells reacting to different stimuli in chemical isolation due to each droplet acting as an isolated microreactor. Furthermore, at high flow rates, the droplets allow massive parallelization, further increasing the throughput of droplets. However, this novel methodology poses unique challenges related to commonly used fluorometry and fluorescent microscopy techniques. We review the optical sensor technology and light sources applicable to DMFC, as well as analyze the challenges and advantages of each option, primarily focusing on electronics. An analysis of low-cost and/or sufficiently compact systems that can be incorporated into portable devices is also presented.

## 1. Introduction

Microfluidics is today a rapidly increasingly active research field due to numerous advantages over batch chemistry and benchtop instrumentation, especially for the implementation of miniaturized, automated analytical and diagnostic devices [1,2,3]. To analyze the sample, microfluidic devices consist of four main components (Figure 1): microfluidic chip, detection, power supply, and communication. The microfluidic section itself can be divided into multiple different subsections (separation, mixing, focusing, droplet generation, etc.) that take care of sample preparation. Due to the small particle and volume size, precise and highly sensitive sensors are used. The most widely used detection method is optical, but electrochemical and mechanical methods also exist. The detected information is either analyzed on the device itself (e.g., smartphone-based devices) or the data is communicated to separate devices (e.g., personal computer). For additional information about detection methods and power supplies, please see the following reviews [4,5].

Droplet microfluidics, as a subfield of microfluidics, is particularly active as it allows analyzing biological organisms, e.g., cells in chemical isolation, enabling more complex assays and/or higher throughput than state-of-the-art methods [6,7,8,9]. In essence, each droplet acts as a separate microreactor, allowing massive parallelization of different reactions and analyses with different types of cells and reagents. Droplet microfluidics relies on two-phase flows of immiscible phases [10], typically oil and water, one of which is discontinuous and forms droplets. Highly monodisperse droplets with <2% coefficient of variation (CV) in size can be generated at frequencies higher than 10 kHz, providing a tool for high-throughput, isolated flow cytometry assays [6,11]. One of the key challenges related to droplet microflow cytometry (DMFC) is the throughput of the sensor: high-throughput analysis is possible only with sensors that have a readout speed the same or higher than the droplet generation frequency. Typically, flow cytometry is conducted by fluorescence spectroscopy that combines laser excitation of fluorophores with photomultiplier tubes for measuring emissions. This yields a high-throughput and highly sensitive system at the expense of physical dimensions and cost, as shown in [9,12,13,14,15]. To make a compact, portable system at a lower price point than widely used benchtop fluorescent flow cytometers, alternative technologies are preferable. However, with more compact alternatives, additional challenges and risks related to (i) signal-to-noise ratio (SNR), (ii) sensitivity, and (iii) spatial and temporal resolution need to be addressed.

Commonly used sensors in microfluidic applications (both experimental and commercial) rely on optical [5] or electrochemical sensing methods [17,18]. Electrochemical sensors (impedance sensors with coplanar or parallel electrode layouts) can be made compact and low-cost, but have limited spatial resolution and issues with SNR [19]. High-throughput flow cytometry typically relies on fluorometry for cell counting and fluorescent microscopy for imaging analyses, e.g., [20]. For both methods, the commonly used setups include (i) lasers as excitation sources, (ii) an optical path consisting of waveguides, mirrors, lenses, etc., and (iii) photomultiplier tubes (PMTs) as detectors. While these setups are reliable, fast, sensitive, and have a good spatiotemporal resolution (including imaging applications), they are also complex and expensive (typically ≥100 k€) [21,22,23]. Therefore, there is a need for research focusing on low-cost, simpler, and more compact detection setups to make measurement setups more available for a wider population and/or speed up the analysis process [24]. For example, relying on light-emitting diodes (LEDs) as excitation sources and photodiodes or Complementary Metal Oxide Semiconductor (CMOS) cameras as detectors, and at the same time including fewer low-cost components in the optical path. While these systems need improvements in sensitivity to be comparable to laser-PMT setups, they can be (i) low-cost, (ii) have small physical dimensions, and (iii) can easily be multiplexed to greatly increase their throughput [25]. Thus, optical systems can be made competitive compared to electrochemical sensors in complexity, dimensions, and cost.

To date, several reviews have focused on biosensors and detection methods in general, as well as on electrochemical and optical sensors in particular, to be applied in Lab-on-a-Chip systems. For instance, in [14,26], high-throughput imaging microflow cytometry is discussed. In [27] detection techniques applicable to droplet microfluidics, including electrochemical and spectroscopic means along with optical ones are reviewed. In [5] optical sensor technology is reviewed and compared to electrochemical and mechanical sensors. In [28] the technology behind optofluidic microflow cytometry is detailed (this method combines micro-optical and microfluidic components). In [29] the focus is primarily on sensor technology, providing a detailed overview of sensor structure, performance characteristics, and limitations from an electronics perspective. Our understanding is that previous reviews can be grouped into two categories: (1) reviews that focus on the analytical performance of experimental setups, and (2) reviews that focus on the specific electronics aspects of sensor technology. Papers in category (1) analyze analytical performance metrics (limit of detection (LOD), sensitivity, analysis speed, sample labelling, etc.). They also discuss the biological and chemical aspects of the experimental setup (types of biological organisms studied, reagents used). These papers only briefly discuss the sensor setups used and the electronics aspects are not discussed in detail. A few review papers cover microflow cytometry or droplet microfluidics, but rarely the combination of the two. Papers in category (2) discuss the details of the electronic sensor technology in general but do not discuss how they are applied in experimental setups in microfluidics or microflow and droplet microflow cytometry. While optical sensor technology is discussed in several papers, light sources and the construction of the optical path are typically not discussed in detail in either category. There is no review giving a balanced overview of the electronic side of optical detection for droplet microflow cytometry.

In contrast to the above, in our review we (i) discuss sensor and light source technology, focusing on electronics, and (ii) discuss how they are applied in experimental setups to detect and analyze droplets, which may contain (a) a set of reagents, (b) cells, or (c) combinations of cells and reagents. We do not discuss the analytical performance, but cover the throughput and characterize the advantages and challenges related to each discussed technology, in light of how they are used in existing experimental setups.

In Table 1, we compare how much detail is provided in the aforementioned review papers on various aspects, and how our paper is positioned compared to them.

The rest of the paper is divided into four main sections. In Section 2, we give an overview of commonly used optical paths in DMFC detection setups. In Section 3, we review the light sources used in DMFC setups and discuss their advantages and challenges related to DMFC detection setup. In Section 4, we review the sensor technology used in DMFC detection setups, in the context of the detection setup as well as in terms of electronics. Finally, in Section 4, we summarize our findings and any potential commercial devices, and we outline the remaining open challenges.

## 2. Light Sources

The focus of this review is on sensors for DMFC; however, a brief look at light sources and optical paths first provides well-needed additional information to better understand the construction of a measurement setup and the choice of the sensor type. Flow cytometry in general and DMFC in particular, require a light source to illuminate the fluorophores in the sample with an appropriate wavelength range and intensity [27]. After conducting a Boolean search in Google Scholar, we concluded that lasers were the most commonly used light sources with about 287 search results. A laser is a light source that produces monochromatic, coherent, and unidirectional light [32,33,34,35,36,37,38,39,40], making it excellent for single-wavelength excitation and thus for DMFC. In this review, we do not focus on the detailed properties and working principles of lasers. For more detailed information about lasers see [35,41]. For benchtop flow cytometers, argon-ion (gas) lasers are commonly used (488 nm wavelength), but their driving circuitry is complex and large [14,37,42]. To overcome this, solid-state lasers, especially semiconductor lasers, can be used. They are lower cost, smaller, and have less complex driving circuits, which makes them more suitable for low-cost and portable applications, as shown in [13,14,38,42,43]. Due to their monochromatic, coherent, and unidirectional light, lasers are suitable light sources for laser-induced fluorescence (LIF) detection in DMFC. With high-throughput, droplets are excited with light for only a fraction of a second. The intensity of the emitted light from fluorescence depends on the intensity of the incident light. The laser beam is guided to the microfluidic chip through an optical fiber, and a set of dichroic mirrors, filters, and lenses to filter and focus the emitted light into 1–3 photomultiplier tubes (PMT) and will lose some of its optical power due to optical parts in the optical pathway. The PMTs convert the light into an electrical signal for the detection of fluorescent events (Figure 2a). This configuration typically results in a high-throughput benchtop instrument. While an overwhelming majority of papers use this approach, this configuration has a lower potential for massive parallelization (i.e., of readout zones, increasing throughput) and portability [14,26,42,43,44,45,46,47]. However, lasers themselves enable focusing high-intensity light beams to fast-moving droplets. Therefore, they are widely used in DMFC applications. For example, the current and future trends for lasers in flow cytometry are reviewed in [48].

Although lasers provide excellent SNR for optical detection, they are not always the optimal technology. In recent years, light-emitting diodes (LEDs) are more commonly used instead. After conducting a Boolean search in Google Scholar, we concluded that LEDs were the second most commonly used light sources with about 136 search results. A low-cost, compact setup can be achieved by using an LED for excitation, with a set of filters and lenses before and after the microfluidic chip, and a photodetector or a camera (e.g., with a CMOS sensor) for detection (Figure 2c). LEDs are the most energy-efficient light sources on the market today [49]. Furthermore, they are compact and capable of producing monochromatic light between the UV (240 nm) and mid-wave infrared (5 µm) ranges, and the light output increases as the technology advances [42,48,50,51]. This makes them suitable as a replacement for xenon arc and halogen lamps [52]. Due to their low energy consumption, LEDs are widely used in handheld instrumentation [53]. In recent years, LEDs have found more use in DMFC applications as they enable decreasing the overall price and size of the measurement device [54]. Furthermore, as LEDs cover the full visible spectrum, they enable matching the absorption wavelength more closely to the fluorophore to achieve maximum excitation efficiency [54]. On the other hand, compared to lasers, LEDs have some disadvantages: the light of the LED is non-collimated, which makes it difficult to focus on the microfluidic channel, whereas lasers usually have collimated light output with spot diameters in the range of a few millimeters. However, this can be overcome by using a lens between the light and the microfluidic channel. The spectrum of the LEDs is narrow and is best described by the manufacturer specification of full width at half maximum (FWHM). Their FWHM is usually in the range of 20–70 nm [55], [56]. Lasers have FWHM in the range of 5–10 nm [57]. Depending on the fluorophore used, this may necessitate additional filters [54]. Compared to lasers, LEDs are less susceptible to overcurrent and simple current regulation circuits are sufficient for driving circuits. Usually, simple resistor-based current limiting circuits are used for low-power LEDs [58] but for higher driving currents a switch-mode constant current driver is more suitable [58,59]. The typical lifespan for LEDs is 50,000 h, and depending on operating conditions, at least 20,000 h [60]. All of these properties make LEDs highly attractive to implement DMFC applications.

For fluorescent imaging and fluorescent microscopy, setups commonly rely on existing microscopy equipment (Figure 2b) or modified versions thereof. This means that the light source will most commonly be a lamp, e.g., a mercury short-arc lamp [61]. Such lamps have a wide emission spectrum, short lifetime [62,63], and may need filtering (e.g., an ultraviolet (UV) filter) [62]. Filtering is specifically needed to reduce the UV emissions harmful to living organisms [62]. They are most suitable for wide spectrum excitation, but due to the wide emission spectrum, monochromators and filters are required to select the appropriate excitation wavelengths [52,53,64]. Compared to lasers or LEDs, the power source is usually high voltage [65,66]. Minimally, the setup needs to include an objective lens and a mirror beside the filters, to direct and focus the light beam into the camera for detection [67]. Alternatively, a laser or high-power LED can be used for excitation, using an objective and a set of filters and optionally additional lenses to filter and focus the light into the camera for detection [26,68]. The popularity of lasers and LEDs is likely due to their long lifetime, easy handling, and inherently monochromatic light beam output.

## 3. Detection Setups and Optical Sensor Technology

Detection in flow cytometry typically relies on optical sensors, primarily fluorescence-based detection methods [73]. In this section, we first review the technology behind the detection setups demonstrated in DMFC applications, and then we discuss the sensors themselves from an electronics perspective. In terms of performance as detection setups, we analyze and compare the throughput of different setups with a specific focus on novel, more compact, and portable setups that can offer similar performance to their widely used, highly sensitive, but large and expensive counterparts. In the case of droplet microfluidic examples are not available for a particular technology, we instead discuss setups using regular microflow cytometry as the optical sensor technology and the construction of the detection setup does not differ (droplets are larger in diameter than individual cells, and are thus easier to detect). Our analysis covers the following aspects: (1) sensor technology, (2) layout of the optical detection setup, (3) droplet counting/imaging throughput. Table 2 summarizes the findings reported in this section and provides a comparison of the performance metrics and setups reported in the literature. Section 4.1 provides an overview of the detection setups used in DMFC and Section 4.2 characterizes and compares the optical sensors available to DMFC.

### 3.1. Detection Setups

Relevant optical sensors can be divided into two major groups: imaging and non-imaging. Imaging sensors can natively record the morphology besides the emitted fluorescent light intensity, and thus are easily applicable to fluorescent microscopy, whereas non-imaging sensors only detect the emitted light intensity and by themselves cannot be used to construct a two-dimensional image. In the group of imaging sensors, there are two major sensor types: Charge-Coupled Device (CCD) and Complementary Metal Oxide Semiconductor (CMOS). For non-imaging optical sensors, there are two major groups: photodiodes and photomultiplier tubes (PMTs). Subtypes exist for both groups. Figure 3 shows a classification chart of the different optical sensors discussed in this section.

To determine which sensors have been most commonly used, we again conducted a Boolean search using Google Scholar. Search results are shown in Figure 4 and are overlaid by the maximum throughput of each sensor to compare popularity with performance. The search indicated that CCD sensors were the most popular (128 results), while PMT sensors came second (101 results). The popularity of CCD sensors is likely because most fluorescent microscopes integrate well with CCD cameras and indicate that most reported setups were used for imaging applications. The relatively high popularity of PMTs was likely due to their high light sensitivity, as is further discussed below. In the following analysis, we discuss the performance of each sensor in more detail. We analyze the performance in terms of (1) quantum efficiency, (2) response time, (3) resolution (spatial/temporal, where applicable), and (4) spectral response. Quantum efficiency (QE) is an essential performance metric of optical sensors, as it expresses the ratio of incident photons to generated electrons [86,87].

CCDs are popular choices for droplet microfluidic devices due to their high light sensitivity, as indicated in [76,88,89,90]. Although CCD sensors are widely used in DMFC, they are not ideal for high-throughput applications. The readout noise for CCD sensors is low, but the maximum frames per second (fps) is limited, which in turn limits the throughput to 100–1000 droplets/s in imaging applications [81,91].

CMOS cameras have 10 times higher framerates than CCD cameras, and therefore are more suitable for high-throughput imaging [91]. In DMFC, CMOS cameras are often used to detect the morphology and textural information of individual cells [14]. Furthermore, CMOS cameras are excellent for massively parallelized applications due to their high spatial resolution and high imaging throughput. The throughput can be increased further by microfluidic channel splitting. Besides their ultrahigh-throughput, these setups were also among the most compact.

PMTs are the most sensitive detectors available for DMFC and are also the most common detectors for high-throughput cell counting applications [31,92,93]. However, they cannot natively resolve 2D images and are fragile and large, which makes them difficult to integrate with a microchannel. Thus, a complex optical path with lenses, filters, waveguides, optical fibers, etc., is needed to direct and focus the light to the microchannel. Furthermore, they can only detect a single color. To detect multiple colors, typically multiple sensors and filters are used, which makes the setup complex and expensive. The throughput of PMT-based non-imaging setups can easily go up to 100,000 events/s in fluorescent event counting applications.

Avalanche photodiodes (APD) can be used to construct highly sensitive, yet more compact and less complex detection setups than those with PMTs; a laser for excitation, an APD, a microscope objective (both for focusing excitation and collecting emissions), and two mirrors [74,94].

### 3.2. Charge-Coupled Device Based Sensors in Droplet Microflow Cytometry (DMFC)

Fluorescence-based detection is most frequently used in conjunction with droplets [31]. Microfluidic chip channel widths are in the range of 50–100 micrometers, and to focus on fluorescent emissions, a lens system is needed, as shown in [95]. Based on the Boolean analysis conducted earlier in Section 2, the CCD sensor is the most widely used sensor in droplet microfluidics. When referred to as a sensor, it is either a camera with a CCD sensor or a standalone CCD sensor with additional acquisition electronics. To capture the emission spectrum from a microfluidic channel, a microscope objective or a set of lenses together with filters and dichroic mirrors are used to filter and focus the emitted light on the sensor [95]. In this review, we do not go into detail on CCD sensor technology, as numerous publications have been published on that subject. Secondly, the state-of-the-art of sensor technology is proprietary to manufacturers and little or no information is present about the latest technologies. More detailed information about CCD technologies is available in [96].

In a CCD sensor, there is an array of biased P-Channel Metal Oxide Semiconductor (PMOS) or N-Channel Metal Oxide Semiconductor (NMOS) photodiodes, each acting as an individual pixel of the sensor. When photons hit the biased photodiode, the photons are turned into an electrical charge. For an array of pixels, there are only a few readout amplifiers, and here lies one of the biggest shortcomings of standard CCD technology in terms of high-throughput droplet analysis. The low number of amplifiers per sensor limits the maximum frames per second (fps) the sensor can achieve [97]. Moreover, the sensitivity of the sensor is limited to the charge-to voltage conversion process, and the readout noise increases if the data is acquired faster [98]. Thus, the readout rate is lowered to minimize the noise [99].

In addition to CCD, intensified CCD sensor (ICCD) and electron multiplication CCD (EMCCD) technologies are used that offer light sensitivity down to a single photon level [100,101]. ICCD sensors have image intensifiers in front of the sensor to boost the number of incoming photons [102]. This improves sensitivity in low-light scenarios at the cost of a higher supply voltage (1 kV) and reduced dynamic range [103,104]. EMCCD sensors have a similar gain performance to ICCD. Instead of the intensifier, an on-chip electron-multiplier is used to achieve the gain [98]. EMCCD has good sensitivity in poor lighting, has little dark current, and better readout noise than ICCD, but also inherits noise from the amplification registry and clock-induced charge [98,99]. A comparison of the noise performance of ICCD, CCD, and EMCCD sensors is presented in [97]. CCD sensors are generally characterized by higher light sensitivity than CMOS sensors, at the cost of imaging throughput. ICCD and EMCCD sensors perform even better in low-light situations [105,106,107,108], but cost more and consume more power. Figure 5 shows the common detection setups for CCD sensor-based measurement devices.

In one demonstrated example, a 488 nm laser was used for excitation and an EMCCD sensor for detection [39]. Droplets of about 350 pL volume were detected in a polydimethylsioxane (PDMS) chip at about 40 Hz droplet generation frequency. Using an LED strobe-light excitation at variable frequency, it was possible to detect droplets at 1150 Hz frequency without the need for a trigger or a synchronizer [75]. Another similar setup was reported in [109] where a 488 nm laser was used for excitation and a camera with an EMCCD sensor was used as a detector. Microdroplets filled with fluorescence were generated at a rate of 30 Hz. When compared to CMOS-based detection setups, the throughput is the most lacking aspect.

### 3.3. Complementary Metal Oxide Semiconductor (CMOS) Based Sensors in DMFC

CMOS sensors are active pixel sensors, as the captured photons are converted to an electrical voltage by photodiodes and amplified in the pixel itself [111,112]. This improves the detection speed at the cost of losing the detection area and sensitivity. Additionally, the pixel fill factor (PFF) can be increased and microlenses can be used [113,114,115]. Compared to CCD sensors, CMOS sensors are typically lower cost, offer lower power consumption, and require lower input voltages [112,116]. Thus, CMOS sensors are more suitable for compact or portable applications, as demonstrated by the literature analysis in the first half of this section. Although CCD sensors have higher light sensitivity, they have a much faster conversion characteristic, making them more suited for high-throughput imaging applications [117]. Sensitivity can be increased by external filtering and focusing or increasing the excitation light intensity. Beyond a certain droplet generation rate or flow rate, motion blur will occur. This can be compensated by increasing the imaging throughput (framerate) of the sensor. However, this reduces the exposure time and therefore the sensitivity, so a more sensitive sensor will be needed.

For high-throughput applications, CMOS sensors are more suitable. From the scientific literature, many high-throughput applications can be found. For instance, a zone-plate array of 64 output channels was demonstrated, capable of counting cells at 184,000 droplets/s throughputs by running an sCMOS camera at 16,000 fps [80]. In another demonstrated setup, the camera and the chip were integrated [72]. By spin coating a filter onto the CMOS sensor and bonding a 16-channel PDMS droplet generator chip, a 100,000 events/s detection rate was achieved. For excitation, a 250 mW LED with 490 nm peak wavelength was used. The filter blocked most of the excitation light and only a 4-pixel wide area of the sensor aligned with the chip was used for detection. The CMOS camera was run at up to 2150 fps. Image stabilization by optomechanical means could also improve the throughput: in one demonstrated setup, a polygon scanner counteracted the movement of a cell in the measured channel. This technique allowed a 1000 times increase in exposure and was suitable for applications where the fluorescent emission intensity was low, as shown in [81]. Figure 6 shows the CMOS-based setups with the highest reported throughput.

Due to the advances in the smartphone industry and specifically smartphone cameras, extremely compact optical paths can be fabricated from low-cost components. Furthermore, it is possible to use an existing smartphone camera with its built-in lens system. One has only to add filters to restrict emissions to the required wavelengths. This setup, using an aperture, can adjust the focal length and focus, as it is shown in [14]. The exact number of filters and lenses may vary from paper to paper, as can be seen in [14,25]. The described setup has the highest potential for physically parallel realization and system-level integration in low-cost, portable instruments because the readout area can be extended by using multiple readers, and the microfluidic throughput can be increased by channel splitting, as shown in [25]. Smartphones have high-performance CMOS cameras, which makes them excellent candidates for use in droplet microfluidics applications due to their low-cost and portability, e.g., [75,90,91,95,118]. Recently, smartphone-based flow cytometry has reached a level where high-throughput can be achieved with low-cost microfluidic setups, as shown in [119]. The solution offers a similar resolution to benchtop microscopes commonly used for droplet analyses and microflow cytometry [120]. Furthermore, they can run software applications that automate analytical workflows and evaluation of results [119,120]. In [77], a theoretical maximum fluorescent event detection rate of up to 1,000,000 events/s was reported using a smartphone camera. In this setup, an ultra-bright LED was flashed in a pseudorandom sequence to excite droplets that would have otherwise overlapped. The system also used a massively parallelized droplet generator structure with 120 channels.

### 3.4. Photomultiplier Tube (PMT)-Based Sensors in DMFC

A PMT is a vacuum tube with a window that consists of a photocathode, an electron-multiplier or dynode, focusing electrodes, and an anode that outputs a current proportional to the incident light [86,121]. The QE of PMTs, defined as the ratio of photoelectrons emitted by the photocathode to the number of incident photons on the window, is usually ~35% [122]. PMTs have response times in the range of nanoseconds, e.g., 26 ns for the Hamamatsu R7205-01. Microchannel plate photomultiplier tubes (MCP-PMT) are advanced PMTs where dynodes are replaced with microchannels of 6–20 µm diameter, decreasing the response time to the picosecond range (e.g., the Hamamatsu R3809U50 has a 0.55 ns response time) and increasing gain to 10^4^–10^7^, while allowing 2D images to be reconstructed [86,123,124]. This comes at the cost of a higher supply voltage (up to 3 kV compared to 0.5–2 kV for a regular PMT). PMTs have a lower power efficiency than CCD and CMOS sensors, require a high voltage power supply (which means a complex power supply unit), are sensitive to magnetic fields, require heating up before operation (takes 30–60 min), and are difficult to handle due to their fragility [125]. Furthermore, due to their high sensitivity, they require a shielding or dark box to operate, adding to the size and complexity [86]. Finally, the performance of PMTs degrades over time: it was found with MCP-PMT that after 5 months of operation, QE dropped by 16% and gain by 50% [126].

For non-imaging sensors, PMTs that have inherently high gain are used, which makes them able to detect fluorescence signals that are weak and have a short lifetime. For many commercial flow cytometers, the PMTs are also used as a sensor (e.g., two widely used BD Accuri C6 and Attune NxT). Multi-parameter measurements have been a challenge with PMTs. To overcome this deficiency, multiple lasers can be switched on separately, varying the excitation and detection wavelength without the use of filters or multiple sensors [84] or single-sensor setups can be used by modulating the laser frequency and using frequency-division multiplexing [83]. Both the aforementioned setups included only the PMT, lasers, optical fibers, and microfluidic chips in their optical path, which is the minimum number of parts achievable with PMT-based setups. Figure 7 shows the common detection setups for PMT-based measurement devices.

### 3.5. Photodiode-Based Sensors in DMFC

In recent years, more versatile and lower-cost silicon-based counterparts, e.g., avalanche photodiodes (APDs), are replacing PMTs. Photodiodes are semiconductor devices that directly convert photons into electrical current. Avalanche photodiodes (APD) are the most closely comparable in performance to the PMTs. They are high-speed and high-sensitivity photodiodes that have internal photocurrent amplification. APDs are physically more robust than PMTs, but still require a higher operating voltage in the range of a few hundred volts, which makes them unsuitable for portable applications [127,128,129]. APDs are sensitive to high ambient temperatures: in one study, a gain reduction of 15% was observed when the sensor temperature increased from room temperature to 80 °C [128]. To overcome that, APD modules with internal temperature compensation circuits might be more suitable for DMFC. Hamamatsu offers multiple modules that have an internal high voltage generator with temperature monitoring and compensation, e.g., the C12702 series [128]. When the diode is operated above the breakdown voltage, it is in Geiger-mode (GM-APD), where it can detect light down to a single photon level [130]. However, due to the avalanche process, the output is not proportional to the incident light. To overcome that, multi-pixel photon counters (MPPC) or silicon photomultipliers (SiPM) were created. In a SiPM device, an array of micro-cells consisting of GM-APD diodes in parallel sums the signal of all cells [130]. The output of SiPM sensors depends on the selected supply voltage that is in the range of 30–60 V [103]. Increasing the supply voltage increases the gain, but also increases the dark count, crosstalk, and after-pulses, which all lower the SNR [130,131,132,133].

In one demonstration, an argon-ion laser was used along with two sets of dichroic mirrors and filters and two APDs to detect two fluorescent signals at a 50 Hz droplet generation rate [134]. It is also possible to combine CCD cameras and APDs to perform rapid kinetic measurements [94]. In another experiment, APD was used for fluorescence emission detection to detect bacteria growth. Additionally, a CCD camera was used to verify droplet generation [74]. Usually, fluorescence is used to label cells, but this can lead to cytotoxicity, nonspecific binding, and other problems. In some cases, high-throughput measurement setups have been provided to measure live cells at a high-throughput rate, using a photodiode as the detector [135,136,137]. Figure 8 shows the common detection setups for the photodiode-based measurement setup.

## 4. Discussion

In this paper, we reviewed the light sources, optical paths (Section 2), and optical sensor technologies (Section 3) applied in the DMFC detection setup. The technology review was focused on the electronics aspect of sensors and light sources and the technology aspects (construction) of detection setups. We focused on fluorometry or fluorescent microscopy as the detection method. In this discussion section, we summarize the findings of Section 2 and Section 3, then highlight existing commercial products, and finally highlight perspectives. The summary combines findings from all previous sections and groups them by the type of detection setup.

Fluorescent counting and microscopy are the leading applications of DMFC technology, and thus setups can be divided into two distinct groups: non-imaging and imaging. Non-imaging detection setups will typically employ lasers as light sources and PMTs as sensors to maximize light sensitivity. This approach requires a highly complex optical path with specialized components, a minimally objective or equivalent lens system, dichroic mirrors, emission filter, and lens (typically 5–10 components). The fluorescent event counting throughput of PMT-based sensors is commonly in the range of 100,000 events per second (eps). This is achieved by fine-tuning the optical path to improve sensitivity. Furthermore, more compact, potentially lower-cost setups can be constructed by using APDs and semiconductor diode lasers while retaining a similar sensitivity. The light sensitivity of PMTs has been achieved thanks to their inherently high gain (10^4^–10^7^). They also have a fast response time (nanosecond–picosecond range). The smaller size and lower input voltage requirements of SiPM sensors can also offer gains up to 10^6^. It is possible to construct more compact yet highly sensitive setups with LEDs and photodiodes, e.g., APDs.

Fluorescent imaging setups commonly rely on existing technology, that is, a fluorescent microscope. These systems typically come with receptacles for CCD/CMOS cameras and use arc lamps (mercury, xenon, or metal halide) as the light source. They also have built-in objectives for magnification. For event counting applications, UV filters are necessary in case lamps are used. Alternatively, lasers and high-power LEDs (250 mW) are used for focused and highly monochromatic excitation. CCD cameras have inherently higher light sensitivity than CMOS cameras, especially ICCD and EMCCD sensors. The optical path minimally consists of a lamp with a UV filter or a laser/LED as the light source, and an objective with or without additional filtering after the microfluidic chip to filter and focus emissions into the sensor. The light source and sensor can be installed at 90 degrees, or a mirror can be used to reflect emissions towards the sensor from the microfluidic chip. With CCD sensors, the maximum imaging throughput is ~1000 frames per second or droplets per second (fps/dps). CMOS cameras have higher framerates than CCDs, with some setups achieving framerates more than a million frames per second [138,139]. They are also more compact and can be equipped with compact lens systems. Due to their high spatial resolution, they can also scan a wider area and thus allow increasing throughput by parallel readouts of branching microfluidic channels (up to 120 channels reported). This yields ultrahigh throughputs of 100,000–200,000 eps in fluorescent event counting applications, and up to 10,000 dps in imaging and morphology analysis. Detection setups most commonly use lasers in conjunction with CMOS sensors. CMOS sensors are excellent for portable applications, with optical paths reported that had only two components in the optical path (lenses and filter). To increase portability and lower cost, it is also possible to use LEDs as light sources. Such setups can reach throughputs between ~1000–2000 dps. Smartphone cameras are also commonly used in portable setups and can yield similarly high throughputs. Furthermore, they are equipped with integrated lens systems and the image processing can be directly implemented on the smartphone, further reducing dimensions and complexity. CCD sensors and CMOS sensors are both pixel sensors and rely on photodiodes but employ different methods for signal amplification. CCD sensors employ charge shifting and a single amplifier, which results in a more consistent (more noise-free), but slower readout than CMOS sensors. On the contrary, CMOS sensors have amplifiers as a part of each pixel, which results in faster but noisier readouts, thus the difference in light sensitivity. However, as CMOS sensor technology improves, the noise and sensitivity cap is narrowing. Increasing the PFF and employing integrated microfabricated lenses on the CMOS sensor are methods to improve sensitivity. Integrating microlenses has led to a reported 30% sensitivity increase in the visible range.

### 4.1. Commercial Platforms

Some benchtop droplet analyzers are commercially available. The Amnis ImageStreamx MKII can detect up to 5000 cells/s. By using multiple lasers, it can detect up to 12 channels of cellular imagery [54]. They probably use a CCD sensor with Time Delay Integration (TDI) readout technology to increase the throughput and maximize the sensitivity [26,140]. Amnis also offers a scaled-down version, FlowSight, that has a CCD camera and can take 10 simultaneous fluorescent pictures up to 4000 events per second [141]. The OptoReader platform (Elveflow, Paris, France) promises a counting throughput of 100,000 events/s [142]. The system relies on multi-wavelength LED/laser excitation and uses a compact, low-cost digital microscope with up to ×100 magnification for imaging. Although not clearly stated in the documentation, the microscope likely uses a CMOS camera. The system is reported to weigh 10 kg and is considered a candidate for Point-of-Care applications [143]. The Cyto-Mine system (Sphere Fluidics Ltd., Cambridge, UK) is a high-throughput benchtop droplet analyzer. It relies on a 488 nm laser for excitation and a CMOS camera for detection [144]. Their droplet sorter is capable of 300 dps throughput [145]. Droplet digital polymerase chain reaction (PCRs) are also high-throughput droplet-based systems, where nucleic acid samples are partitioned into thousands of droplets. The readout is based on fluorescent event counting, using a laser/LED as a light source and a PMT for detection [146]. A more detailed comparison of four commercially available (Accuri^TM^ C6 (BD Biosciences, San Jose, CA, USA), NovoCyte ^®^ (ACEA Biosciences, San Diego, CA, USA), Attune^TM^ NxT (Thermo Fisher Scientific, Waltham, MA, USA), and MACSQuant 10 (Miltenyi Biotec, Bergish Gladbach, Germany)) cytometers is available here [147]. The flow cytometry buyers guide can also be helpful when selecting a platform [141].

### 4.2. Perspectives

We can conclude that with recent developments in semiconductor sensor technology (photodiodes and CMOS sensors), it is possible to construct high-throughput fluorescent counting and microscopy setups that are on par in performance with well-established benchtop counterparts (PMTs and traditional fluorescent microscopy setups). Using compact detection setups relying on lasers for excitation and CMOS sensors for detection, it is possible to reach counting throughputs above 100,000 eps, and imaging throughputs above 10,000 dps. With even more compact setups that only employ a lens and a filter in conjunction with LEDs for excitation and CMOS sensors for detection, it is possible to reach above 1000 dps. Thus, highly portable and high-throughput imaging and counting setups are achievable. These setups commonly rely on the parallelized readout of branching microfluidic channels with thousands of droplets passing through each.

In the future, we can expect further development in CMOS sensor technology, increasing the sensitivity and decreasing the cost of sensors. With the rapid development of parallelized image processing architectures, the computational overhead will also continue to drop, increasing the throughput of the system further. Image quality can be increased dramatically using machine learning for de-noising and pre-processing. Neural networks can also be taught to detect and classify cells in a completely automated manner. With the cost and power requirements of such systems dropping rapidly, fully automated portable analyzers are on the horizon and can greatly aid in the fight against novel and recurring bacterial pathogens. There is a pronounced need for a high number of portable analyzers to decentralize diagnostics and increase diagnostic coverage. Early detection can greatly aid preventive measures and targeted isolation of cases to prevent community spread. Although there is a significant scientific and commercial interest in portable droplet analyzers, several open challenges remain. To make such detection setups competitive compared to benchtop instruments, CMOS sensors still need to become more sensitive. The bottleneck of the analog-to-digital (ADC) conversion also remains an issue for portable applications. The miniaturization of lenses and filters is an ongoing process, but the highly specialized fabrication methodology required for them is a limit. If the optical path could be fabricated with lower costs, e.g., 3D printing, that would greatly reduce the overall complexity and cost (as well as shorten the supply chain for instrument fabrication).

## Figures and Tables

**Figure 1 micromachines-12-00345-f001:**
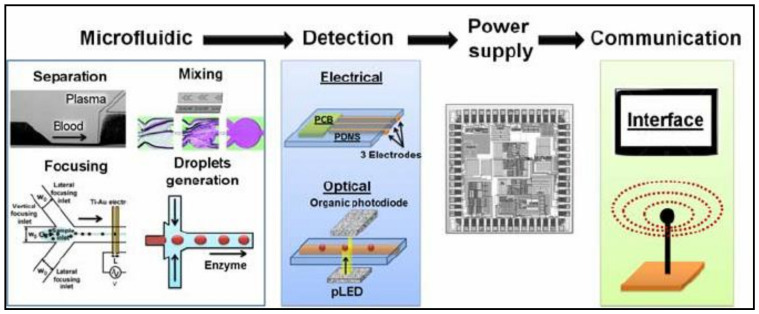
Typical setup of a self-contained microfluidic system that consists of four major sections: microfluidic chip where a type of microfluidic process occurs (mixing, separation, focusing, droplet generation, etc.); detection—optical, electrochemical, or mechanical detection methods are used to detect cells in the microfluidic chip; power supply to power the device; communication—gathered data is either analyzed on the device or communicated to separate device. Reproduced with permission from [16].

**Figure 2 micromachines-12-00345-f002:**
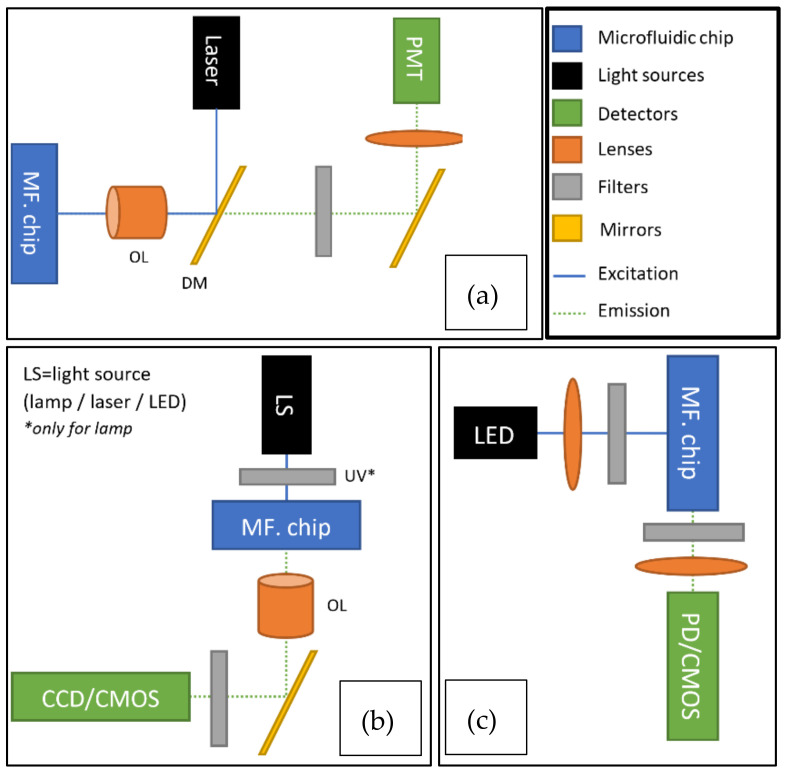
Typical light path configurations applicable in droplet microflow cytometry (DMFC) setups, using lasers, light-emitting diodes (LEDs), and arc discharge lamps as light sources. Setups are fine-tuned by the addition of filters and lenses of various types to focus and filter excitation and emission beams. (**a**) Typical fluorescent event counting setup with a laser as the light source and objective lens (OL) and two dichroic mirrors (DM) to focus and direct fluorescent emission [14,43,44,45,46]. (**b**) Typical fluorescent microscopy setup using a mercury arc or halogen lamp and a set of filters to select the appropriate excitation wavelength [61,69]. Alternatively, laser/LED excitation can be used without excitation filtering [70,71]. (**c**) Compact LED-based fluorescent imaging/microscopy setup [25,72]. Only narrow-band LEDs are suitable for use without excitation filtering. By combining the setups shown in (**a**,**b**), one can increase the spatial or temporal resolution of the imaging system.

**Figure 3 micromachines-12-00345-f003:**
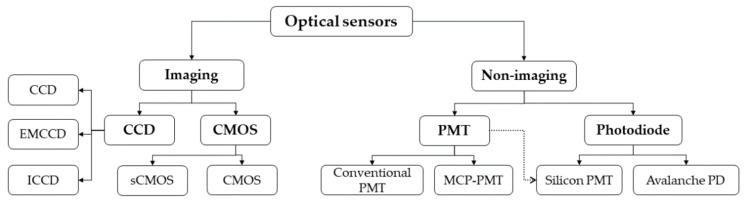
Based on the operating methods, the optical sensors for droplet microfluidic setups can be divided into two major categories—imaging and non-imaging. Photomultiplier tubes or photodiodes are widely used for non-imaging detection setups in droplet microflow cytometry (DMFC) where they detect the light level. If morphological and/or spatial information about cells is required, Charge-Coupled Device (CCD) or Complementary Metal Oxide Semiconductor (CMOS) type imaging sensors are preferred.

**Figure 4 micromachines-12-00345-f004:**
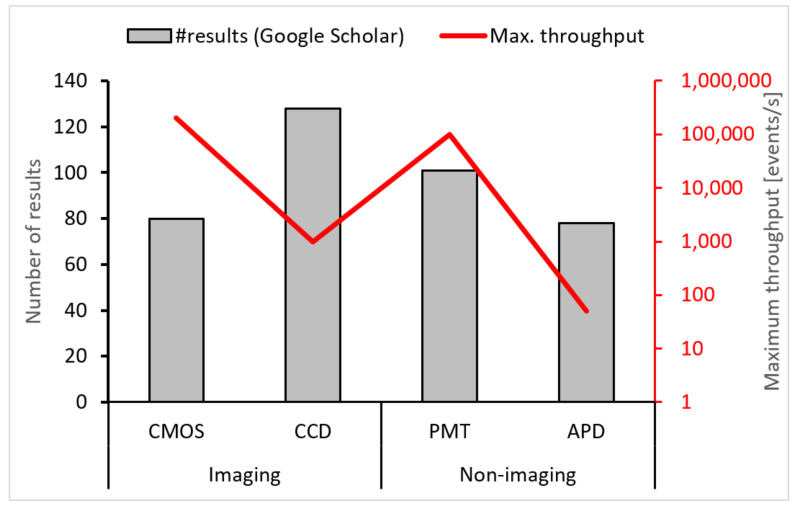
Sensor types and their relevance to droplet microflow cytometry (DMFC). On the vertical axis, there are several results for each sensor type based on a Boolean search from Google Scholar. The red line shows the maximum reported throughput of each sensor type.

**Figure 5 micromachines-12-00345-f005:**
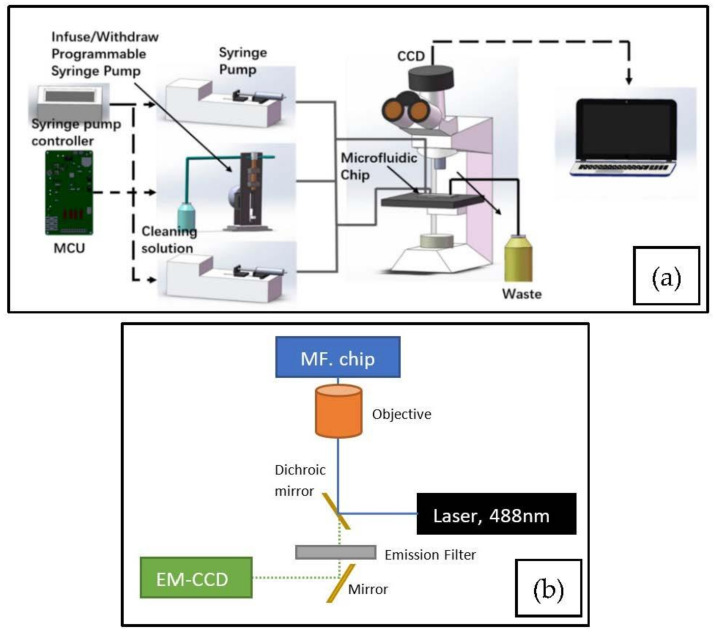
Detection setups used in droplet microflow cytometry (DMFC) using Charge Coupled Device (CCD) sensors as a detector. (**a**) A typical solution for microscope-based microfluidic measurement setup. A microscope with an integrated camera is used to zoom and focus on a microfluidic chip. Two syringe pumps with a controller are responsible for the continuous flow of sheath (carrier) fluid and sample fluid. (**b**) A laser Electron Multiplying Charge Coupled Device (EMCCD) sensor system capable of detecting fluorescence-induced droplets at the rate of 30 Hz. The optical setup consists of a 20× objective lens, dichroic mirror, emission filter, mirror, camera, laser, and a microfluidic device. Reproduced with permission from [110].

**Figure 6 micromachines-12-00345-f006:**
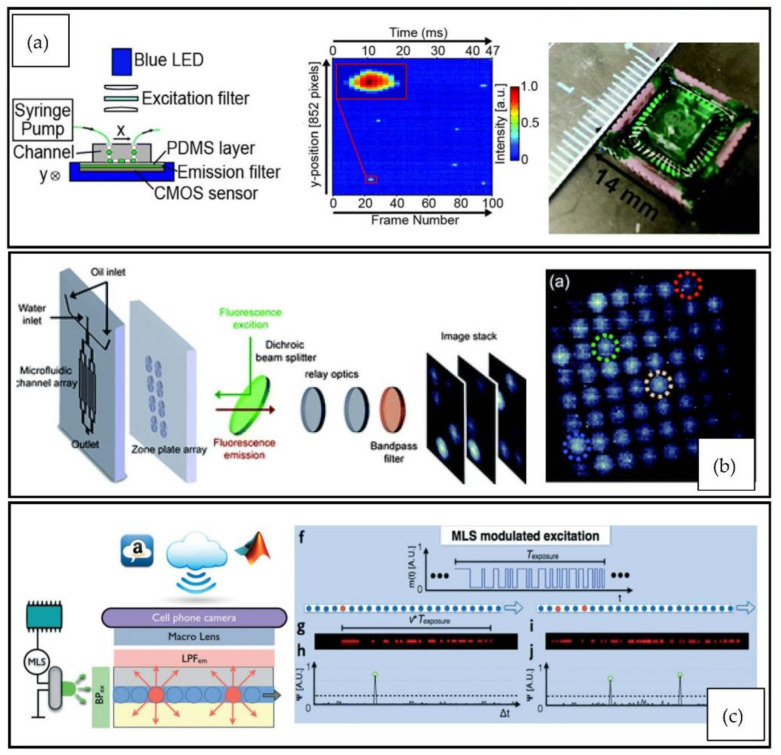
Demonstrated ultrahigh-throughput detection setups in DMFC. (**a**) A compact LED-CMOS system, which could detect fluorescent droplets at 254,000 dps throughput. The system used a simple and compact optical path and microfluidic channels branching into 16 parallel channels to increase throughput [72]. (**b**) A laser-CMOS system, which could detect droplets at 184,000 dps throughput. In this application, microfluidic channels were split into 64 parallel branches and imaged through an 8 × 8 zone-plate array. The resultant image is shown on the right [80]. (**c**) An LED-CMOS system capable of detecting droplets at up to 1,000,000 dps throughput. This was achieved by splitting the microfluidic channels into 120 parallel branches. Additionally, pseudorandom maximum length sequences (MLS) were used for excitation that prevented droplets overlapping due to framerate limitations of CMOS cameras [77]. Reproduced with permission from [72,77,80].

**Figure 7 micromachines-12-00345-f007:**
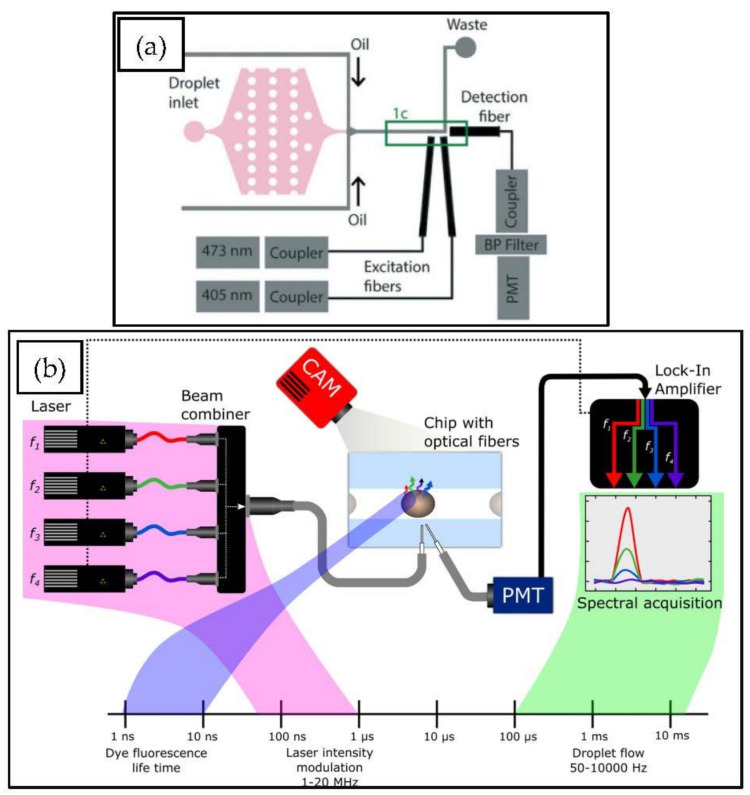
Detection setups used in droplet microflow cytometry (DMFC) using photomultiplier tubes (PMT) as a detector. (**a**) Measurement system where multiple lasers are used and coupled into the microfluidic chip. Knowing the flow speed, multiple analyses of droplets can be done using only one PMT tube [84]; (**b**) a measurement setup consisting of one PMT capable of measuring four parameters at the same time, using lasers for light sources, beam combiner, and lock-in amplifier to demodulate the result [83]. Reproduced from with permission from [83,84].

**Figure 8 micromachines-12-00345-f008:**
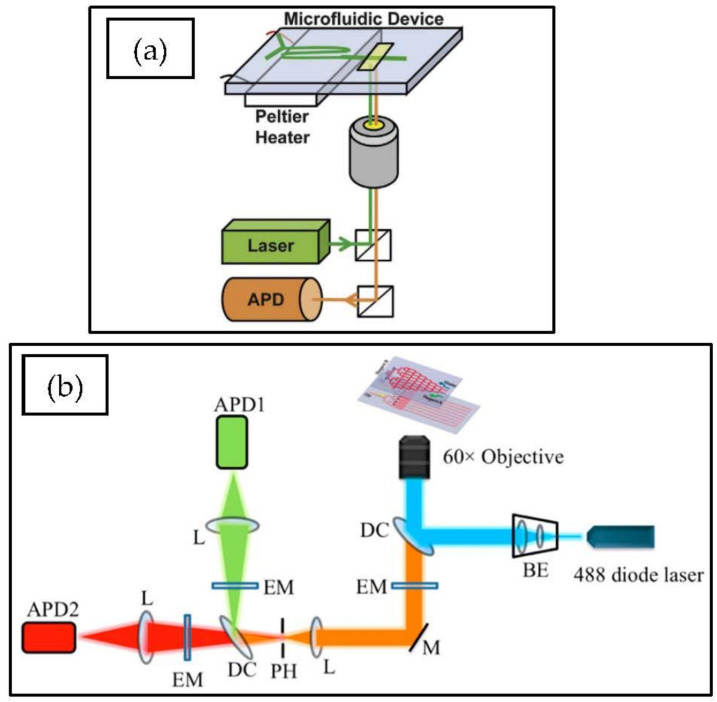
Detection setups used in droplet microflow cytometry (DMFC) using photodiodes or avalanche photodiodes (APD) as a detector. (**a**) Microfluidic measurement system where a laser is used for excitation and APD is used as a detector. Additionally, optics are used to focus light on the sample. (**b**) Microfluidic measurement system where Differential Detection Photothermal Interferometry is used and two photodiodes collect the data that is collected with a lock-in amplifier and analyzed in PC. Reproduced from with permission from [74,134].

**Table 1 micromachines-12-00345-t001:** Comparison of reviews on detection techniques in microflow cytometry. The level of detail of the various aspects discussed is rated from low to high (*–***).

Ref.	Sensor Technology	Light Source Technology	Optical Path Construction	Analytical Performance	Droplet Microfluidics?
[14]	**	*	***	**	No
[5]	**	*	***	**	No
[26]	**	**	**	*	No
[27]	**	*	***	***	Yes
[28]	**	**	**	**	No
[30]	*	*	*	**	Yes
[31]	*	**	*	**	Yes
Our paper	*******	******	**	*	Yes

**Table 2 micromachines-12-00345-t002:** Comparison of the complexity and performance of detection setups used in droplet microflow cytometry (DMFC) systems.

Optical Sensor	Light Source	Max. Throughput (Dps) *	Excitation Wavelength (nm)	Complexity (No. of Optical Components **)	Portable/Compact?	Imaging?	Ref.
APD	laser	50	488	>10	no	no	[74]
CCD	LED	1150	~440	4	no	yes	[75]
CCD	lamp	100	470–495	>10	no	yes	[76]
EM-CCD	laser	40	488	6	no	yes	[71]
CMOS	LED	1,000,000	530	3	yes	no	[77]
CMOS	LED	254,000	490	3	yes	no	[72]
CMOS	laser/LED	96,000	488/640	>10	no	yes	[78]
CMOS	laser	70	532	2	yes	yes	[79]
sCMOS	laser	184,000	532	5	no	no	[80]
sCMOS	laser	10,000	488/560	>10	no	yes	[81]
PMT	laser	100,000	488	5	no	no	[82]
PMT	laser	10,000	405/488/561/639	6	no	no	[83]
PMT	laser	500	488	3	no	no	[43]
PMT	laser	50	405/473	7	no	no	[84]
PMT	laser	10	445	7	no	no	[85]

* dps = droplets or cells per second. ** includes all mirrors, filters, lenses, waveguides, apertures, etc., in the optical path, but not the microfluidic chip, the sensor, nor the excitation source.

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
