# Peer review of "Optical Detection Methods for High-Throughput Fluorescent Droplet Microflow Cytometry"

_micromachines, 2021, doi:10.3390/mi12030345_

Round 1
Reviewer 1 Report
This review paper seems to be very useful in the field of droplet based assay platform. However, some minor issues should be clarified before publication to the Micromachines.
- In table 1 of comparison data of reviews on detection techniques, more refereces of droplet microfluidics might be added in this table. Especially, some references of DMFC should be more suitable.
- The reviewer suggests that chapter of various applications for high-throughput fluorescent DMFC might be included for potential readers in these research fields.
- Other examples of commercial platforms shoud be included in this review for more valuable informations.
Author Response
Dear reviewer,
thank you for your review, input and comment. Please see the file attached where answer to your feedback is given.
Best regards,
Kaiser Pärnamets (on belahf of authors)

Reviewer 2 Report
Comments: The manuscript deals with flow cytometry technologies for droplet microflow cytometry (DMFC). A variety of topics on droplet microflow cytometry are surveyed based on optical sensors and light sources, however, there seem to be several serious issues with the manuscript. My detailed comments and possible approaches to improve the manuscript are mentioned below. Major concerns: 1. Reviewer cannot understand why these is difference in the optical sensor and lights source for droplet and cell flow cytometry? These is only a size difference, the imaging target is the cell or other biological sample in the droplet, the significance of this manuscript is not clear. 2. In introduction a conceptual image for DMFC, and a flowchart of the manuscript should be given for better understanding. 3. The largest difference between droplet and cell flow cytometry is the microfluidic platform. Combination of microfluidic droplet generation and optical imaging or detection should be the major novelty of this manuscript. And more content should be added 4. How the size, structure, components of the dropt influence on the selection of light source and sensors should be given. 5. Lack of recent important references which are possible for DMFC. For example, 1.Intelligent image-activated cell sorting, Cell 175 (1), 266-276. e13,2018 2.Label-free chemical imaging flow cytometry by high-speed multicolor stimulated Raman scattering, Cytometry Part A 91 (5), 494-502 3.High‐throughput, label‐free, single‐cell, microalgal lipid screening by machine‐learning‐equipped optofluidic time‐stretch quantitative phase microscopy, Proceedings of the National Academy of Sciences 116 (32), 15842-15848 4.Optofluidic time-stretch quantitative phase microscopy, Methods 136, 116-125 5.Intelligent whole-blood imaging flow cytometry for simple, rapid, and cost-effective drug-susceptibility testing of leukemia, Lab on a Chip 19 (16), 2688-2698 Other minor concerns must be addressed: 1. The quality and style of those images in this manuscript are poor. 2. These are lots of mistakes, errors, typos, duplicate ref over the entire manuscript.Author Response
Dear reviewer,
thank you for your review, input and comment. Please see the file attached where answer to your feedback is given.
Best regards,
Kaiser Pärnamets (on behalf of authors)

Round 2
Reviewer 2 Report
Authors have addressed all my concerns.
Attached file of the manuscript is not a final version, and the modified sections should be marked red, not just show the corrections log by office.